# Eating Fast Has a Significant Impact on Glycemic Excursion in Healthy Women: Randomized Controlled Cross-Over Trial

**DOI:** 10.3390/nu12092767

**Published:** 2020-09-10

**Authors:** Yuuki Saito, Shizuo Kajiyama, Ayasa Nitta, Takashi Miyawaki, Shinya Matsumoto, Neiko Ozasa, Shintaro Kajiyama, Yoshitaka Hashimoto, Michiaki Fukui, Saeko Imai

**Affiliations:** 1Department of Food and Nutrition, Faculty of Home Economics, Kyoto Women’s University, 35, Kitahiyoshi-cho, Imakumano, Higashiyama-ku, Kyoto 605-8501, Japan; k5231047@yahoo.co.jp (Y.S.); ayasanman.n.vabo7915@gmail.com (A.N.); takashiukb@gmail.com (T.M.); matumots@kyoto-wu.ac.jp (S.M.); 2Kajiyama Clinic, Kyoto Gojyo Clinic Build. 20-1, Higasionnmaeda-cho, Nishinanajyo, Shimogyo-ku, Kyoto 600-8898, Japan; kajiyama-clinic@dream.ocn.ne.jp; 3Graduate School of Medical Science, Kyoto Prefectural University of Medicine, 465 Kajii-cho, Kawaramachi-Hirokoji, Kamigyo-ku, Kyoto 602-8566, Japan; y-hashi@koto.kpu-m.ac.jp (Y.H.); michiaki@koto.kpu-m.ac.jp (M.F.); 4Graduate School of Medicine, Kyoto University, 54, Kawahara-cho, Syogoin, Sakyo-ku, Kyoto 606-8507, Japan; nei126@kuhp.kyoto-u.ac.jp; 5Japanese Red Cross Kyoto Daini Hospital, 355-5, Kamanza, Marutamachi, Kamigyo-ku, Kyoto 602-8026, Japan; kaji20091025abcd@gmail.com

**Keywords:** diet, eating speed, eating fast, glycemic excursion, postprandial glucose, flash glucose monitoring, diabetes, obesity

## Abstract

Epidemiological studies have shown that self-reported fast eating increases the risk of diabetes and obesity. Our aim was to evaluate the acute effect of fast eating on glycemic parameters through conducting a randomized controlled cross-over study with young healthy women. Nineteen healthy women wore a flash glucose monitoring system for 6 days. Each participant consumed identical test meals with a different eating speed of fast eating (10 min) or slow eating (20 min) on the 4th or the 5th day. The daily glycemic parameters were compared between the 2 days. The mean amplitude of glycemic excursion (MAGE; fast eating 3.67 ± 0.31 vs. slow eating 2.67 ± 0.20 mmol/L, *p* < 0.01), incremental glucose peak (IGP; breakfast 2.30 ± 0.19 vs. 1.71 ± 0.12 mmol/L, *p* < 0.01, lunch 4.06 ± 0.33 vs. 3.13 ± 0.28 mmol/L, *p* < 0.01, dinner 3.87 ± 0.38 vs. 2.27 ± 0.27 mmol/L, *p* < 0.001), and incremental area under the curve for glucose of dinner 2 h (IAUC; 256 ± 30 vs. 128 ± 18 mmol/L × min, *p* < 0.001) for fast eating were all significantly higher than those for slow eating. The results suggest that fast eating is associated with higher glycemic excursion in healthy women.

## 1. Introduction

It has been demonstrated that self-reported eating at a fast speed leads to weight gain [1,2], increases risk of type 2 diabetes [3,4,5], obesity [6,7], and metabolic syndrome [8,9,10] in epidemiological and cohort studies. Eating fast encourages higher energy intake by increasing food stimuli, hunger, and the desire to eat [11,12,13]. However, in epidemiological and retrospective reports, eating speed was entirely assessed by self-reported questionnaire, which might lead to report bias and eating speed might lack objectivity, leaving the effect of eating speed on glycemic response unclear. Additionally, an interventional study of the effect of eating speed on glycemic response has not been investigated. Therefore, the rate of eating speed should be evaluated objectively by an interventional randomized control study.

The aim of this study was to evaluate the acute effect of different eating speeds on glycemic parameters in young healthy women with a flash glucose monitoring system (FGM, FreeStyle Libre Pro, Abbott Japan, Tokyo, Japan). FGM is a new technology among continuous glucose monitoring systems that does not require regular capillary glucose sampling by finger prick (self-monitored blood glucose, SMBG) like traditional continuous glucose monitoring (CGM). The FGM sensor stores interstitial fluid glucose levels every 15 min for 14 days and was reported to be accurate and effectively replaced SMBG [14,15,16].

## 2. Materials and Methods

### 2.1. Participants

University students were recruited from Kyoto Women’s University, Kyoto, Japan, after being informed the study requirements. Twenty-one participants were enrolled in the study and 2 participants were excluded because of discontinuation of the study protocol of eating speed. The study was conducted between December 2019 and February 2020. The volunteers had no history of any metabolic diseases. None of the volunteers were pregnant, smokers, had an eating disorder, weight loss, or followed any other special diet in the previous 6 months, and they refrained from taking any medications and supplements known to affect their metabolism. The purpose, design, and risks of this study were explained to each participant and written informed consent was obtained prior to the study.

### 2.2. Study Design

The study was designed as a randomized controlled two-treatment cross-over within-participant clinical study to avoid the characteristic differences of the two groups in different eating speed. The study protocol involving human subjects was approved by the Ethics Committee of Kyoto Women’s University (2019-8) according to the guidelines laid down in the Declaration of Helsinki and was registered at the UMIN Clinical Trials Registry (UMIN 0000038684). All participants consumed identical test meals for two days, which were consumed at two different eating speeds during the 6 day study period, as shown in Figure 1. The study protocol was explained to each participant prior and during the study; the participants were reminded to follow the protocol by phone for thorough eating procedure control. The participants wore FGMs on the back of their left upper arm for 6 days under the physician’s management at Kyoto Women’s University. Although the FGM can start recording the glycemic parameters 1 hour after wearing, to obtain stable and accurate glycemic data, we did not start the interventional study until the 4th day. On the 4th day, participants were instructed to consume three meals either quickly (10 min) or slowly (20 min). On the 5th day, they were instructed to consume each meal at the opposite speed. The order of the two eating speeds was determined randomly prior to the beginning of the study. The eating protocol, the sequence of the dishes consumed, and time required to consume them was defined in slow eating, as vegetables for 7 min, main dish for 7 min, and rice/bread for 6 min (in total of 20 min) [17]. In fast eating, the dishes in the identical meal were mixed to be consumed in a total of 10 min. On the 6th day, FGMs were removed by the participants themselves under the physician’s instruction at Kyoto Women’s University. The data recorded in FGM were extracted and daily glycemic parameters were compared within-participant between 2 days of consuming identical meals at a different eating speed.

### 2.3. Test Meals

Table 1 shows the composition and macronutrient content of the test meal. The frozen boxes of fried fish and vegetable at lunch, and gluten-meat steak and vegetable at dinner were purchased (Tokatsu Foods, Yokohama, Japan) and provided to the participants by the researchers. The rest of the food was prepared by the participants according the brochure prepared by the dietitians. The frozen food boxes for lunch and dinner were kept in the freezer until consumption. Test meals of 200 g of boiled white rice and 90 g of white bread were measured exactly and heated by each participant before consumption. Throughout the study period, the participants were allowed to consume, other than test meals, only water, green tea, tea, and coffee without sugar nor milk. The participants were requested to avoid alcohol and excessive physical activity for 2 days prior to the study and during the study period. Each participant was instructed to follow the study protocol precisely during the study period and the collected records of eating speed and the amounts of food were assessed for compliance of the study protocol by the dietitians of the study group. The dietitians excluded the participants who did not follow the protocol.

### 2.4. Measurements

Two weeks before the study, anthropometric measurements and blood samples of participants were collected in the morning after an overnight fast. Blood samples were examined in Rakuwakai Toji Minami Hospital. Fasting plasma glucose concentration (FPG) was measured by amperometric methods. Hemoglobin A1c (HbA1c) levels were determined by high-performance liquid chromatography (HPLC). The incremental area under the curves (IAUC) for glucose after breakfast, lunch, and dinner were calculated from the baseline by the trapezoidal method. The parameters to evaluate glycemic variability were the mean amplitude of glycemic excursions (MAGE) [18] and the standard deviation (SD) of plasma glucose. These glycemic parameters were compared within-participant for 2 days of consuming the identical meals with different eating speeds.

### 2.5. Sample Size and Statistical Analysis

A sample size of 14 participants in the current study provided 95% power to detect 5% difference in postprandial glucose concentrations (G*Power 3.1, Heinrich-Heine-Universität Düsseldorf, Germany), referring to our previous study of consuming test meals in different sequences in healthy women [19]. Twenty-one participants enrolled in the study. The primary outcome was postprandial glucose concentration and the secondary outcomes were MAGE and IAUC for glucose. We could not confirm normal distribution and homogeneity for all glycemic parameters by Shapiro–Wilk and Levene tests, so we performed a paired comparison by the Wilcoxon matched-pairs signed-rank test, and *p* < 0.05 was considered statistically significant. The results are expressed as mean ± standard error of the mean (SEM) unless otherwise stated. All analyses were performed with SPSS Statistics ver. 22 software (IBM Corp., Armonk, NY, USA). The composition and macronutrient content of the menu of the test meals was shown in Appendix A.

## 3. Results

The results were based on 19 women (20.8 ± 0.6 years, BMI 20.6 ± 1.9 kg/m^2^, HbA1c 34 ± 2 mmol/mol (5.4 ± 0.2%), FPG 4.86 ± 0.39 mmol/L: mean ± SD). Figure 2 demonstrates the postprandial glucose profiles for two different eating speeds in young healthy women. The SD, MAGE, incremental glucose peak (IGP) of breakfast, lunch, and dinner, and IAUC for glucose 2 h after dinner for fast eating (10 min) were all significantly higher compared to those for slow eating (20 min), as described in Table 2. Although, the mean plasma glucose concentration showed no difference between the 2 days of fast and slow eating.

## 4. Discussion

To the best of our knowledge, this is the first interventional study to investigate the association between eating speed and glycemic parameters by FGM. The results of this study suggest the possibility that fast eating induces higher postprandial glucose concentrations and higher daily glycemic excursions in young healthy women. Numerous studies reported that fast eating was associated with increased body weight and overeating [1,2,6,7,11,12,13,20], elevated blood pressure and fasting plasma glucose concentration [8,10], increased insulin resistance [4], and increased the risk of impaired glucose tolerance and type 2 diabetes [3,5]. It has also been pointed out the association between fast eating and lipid abnormality, such as elevated plasma triglyceride and reduced plasma HDL concentration [10,21]. However, it is important to mention that the extent of eating speed in these previous reports was assessed subjectively, such as using questionnaires answered by participants themselves, which might yield bias in evaluating the effect of eating speed on physiological parameters. The interventional method used in the present study is expected to evaluate objectively the effect of eating speed on glycemic parameters.

On the other hand, several reasons may be that eating slowly may exert its beneficial effect by enhancing diet-induced thermogenesis (DIT), increasing serum adiponectin concentration, and suppressing endotoxin and non-esterified fatty acid, as reported previously [22]. It has been reported that interleukin-1β and interleukin-6, which are both involved in insulin resistance, are decreased in individuals with slow eating [23,24], indicating the link between eating speed and glycemic parameters observed in the present study. Moreover, eating slowly may influence gastrointestinal satiety hormones, such as ghrelin and peptide tyrosine-tyrosine (PYY) which control appetite and influence food consumption, suggesting that modifiable eating behaviors actually regulate the hormonal response to food [25]. In contrast, Shah M et al. reported that eating speed could not be explained by the changes in meal-related hormones. In their study, eating breakfast slowly (30 min) and quickly (10 min) did not affect postprandial gut hormone responses such as ghrelin, glucagon-like-peptide-1 (GLP-1), PYY, nor hunger and daily food consumption [26]. In the present study, IAUC for glucose of breakfast and lunch demonstrated no difference, possibly because the secretion of the incretin hormones might not be affected by eating speed in the daytime. Additionally, another possible reason was that the energy ratio of dinner was large (40%) compared to that of breakfast (25%) and of lunch (35%). We employed this energy ratio to design the three test meals according to the general meal plans of Japanese dietary habits. However, as there are controversial reports in association with gastrointestinal hormones and eating speed, the mechanisms underlying the association with eating speed and metabolic responses needs to be verified in further studies.

The strength of our study is that this is the first randomized controlled cross-over interventional study to explore the association between eating speed and the glycemic responses in healthy Japanese women. Postprandial hyperglycemia and higher MAGE are associated with increased risk of type 2 diabetes and cardiovascular diseases in people with and without diabetes [27,28,29]. It is reported that the large blood glucose fluctuation not only increases tumor necrosis factor-α, interleukin-6 [30], and platelet aggregability [31], but also decreases endothelium dependent vasodilation [32] even before the onset of diabetes. Therefore, decreasing postprandial glucose concentrations and MAGE may reduce the risks of developing impaired glucose tolerance, type 2 diabetes, and cardiovascular diseases in people without diabetes.

Some limitations to the present study should be mentioned. First, the present study is an acute interventional study, therefore, it is unable to translate all these effects on glycemic responses to long-term benefits. Second, the participants of this study consisted only of young healthy Japanese women who experience a diet and lifestyle specific to Japan. Therefore, we should be cautious to apply our results to individuals with other gender, race, genetic backgrounds, and lifestyle, and individuals with diabetes. For the third, because how the metabolism is regulated by eating speed is not fully understood, the role of insulin, incretin hormones, cytokines, and endogenous glucose production on significance of eating speed is still unclear. Fourth, the eating protocol between fast eating (eating sequentially) and slow eating (eating at once as a mixture) in this study was not exactly the same, leaving possibility that the difference in the eating protocol contributed to the results. As the fifth limitation, psychological conditions such as satiety may have influenced the glycemic responses, but we did not measure the extent of satiety between the study days of fast and slow eating. It has been reported that eating fast tended to increase the amount of food intake by suppressing satiety [11,12,13]. Since there was no difference in the amount of food taken between study days of fast and slow eating in our study, how and/or whether satiety influenced the glycemic parameters observed in our study is left unanswered. The sixth limitation was the possibility that participants might not have followed strictly the study protocol. Although, we had thoroughly and repeatedly explained and instructed the protocol prior to and during the study period to maintain high adherence. Therefore, further study needs to explore the comparison between fast eating and slow eating on glycemic responses when meals were consumed in the same manner, with the meal sequence of vegetable, main dish, and rice/bread.

The disadvantage of eating fast shown in this study brings the possibility of raising the risk of type 2 diabetes and obesity in healthy young individuals. Eating slowly is potentially advantageous to public health, because the modification of eating speed could be cost effective for promoting management of body weight and glycemic control [33,34]. Dietary education on the benefit of eating slow could be a simple way to reduce excess food intake, since eating fast leads to higher energy intake but lower satiety [11,12,13]. One practical approach that can be done for promoting slow eating behavior is to try eating slowly at lunch breaks in schools or workplaces. However, in future studies, additional investigations are required to explain the mechanisms under these effects and the long-term effects of eating speed on metabolic control in individuals with and without diabetes.

## 5. Conclusions

To summarize, we have shown that eating fast is associated with higher daily glycemic excursions and postprandial glucose concentrations in a randomized controlled cross-over trial for the first time. Real-world approaches are needed to better understand the negative influence of eating fast on glycemic responses and to support approaches for slowing down eating speed in healthy individuals.

## Figures and Tables

**Figure 1 nutrients-12-02767-f001:**
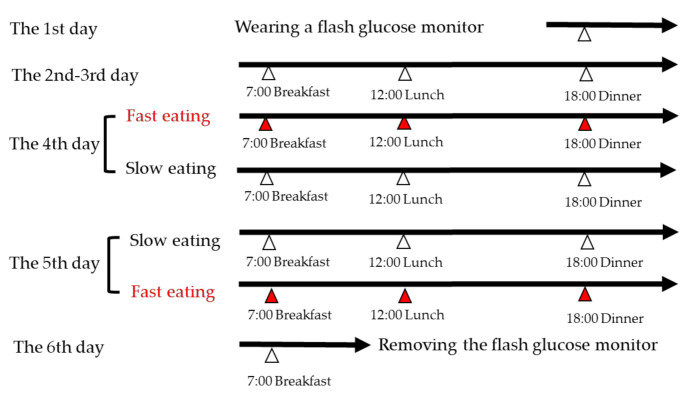
Study protocol. Participants consumed identical test meals for 2 days over the 6-day study period with flash glucose monitors. Participants consumed test meals of breakfast at 07:00, lunch at 12:00, and dinner at 18:00 for fast eating (10 min) or slow eating (20 min) on the 4th or the 5th day in the randomized controlled cross-over study. Red triangle—fast eating; white triangle—slow eating.

**Figure 2 nutrients-12-02767-f002:**
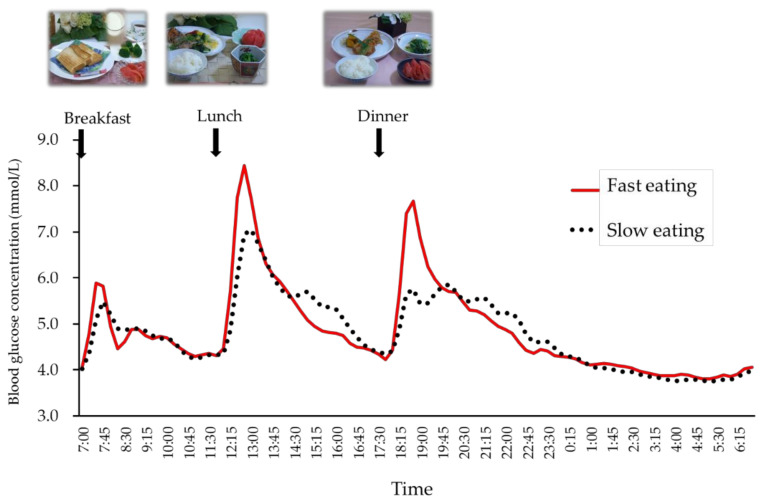
The mean plasma glucose profiles of fast and slow eating in healthy women (*n* = 19). Each participant consumed identical meals for fast eating (10 min) or slow eating (20 min) on the 4th or the 5th day. Red solid line—fast eating; black dotted line—slow eating.

**Table 1 nutrients-12-02767-t001:** The composition and macronutrient content of the test meals.

Meal	Energy	Protein	Fat	Carbohydrate	Fiber	Detail Content
	(kcal)	(g)	(g)	(g)	(g)	
Breakfast	437	18.2	12	70.1	5.8	White bread 90 g, tomato 100 g, broccoli 60 g, milk 200 g, strawberry jam (sugar free) 13 g
Lunch	624	25.1	11.5	104	8.1	Boiled white rice 200 g, frozen lunch box of fried fish with vegetable, tomato 100 g, spinach 80 g
Dinner	689	23.6	17.4	107.6	7.8	Boiled white rice 200 g, tomato 100 g, frozen lunch box of gluten-meat steak with vegetable, spinach 80 g with fried tofu 15 g
Total	1750	66.9	40.9	281.7	21.7	

The macronutrient content of the test meals was calculated by computer software (Microsoft Excel Eiyokun for Windows Ver.7.0, Kenpakusya, Tokyo, Japan).

**Table 2 nutrients-12-02767-t002:** Characteristics of glycemic parameters of fast or slow eating in healthy women (*n* = 19).

Glycemic Parameters	Fast Eating (10 min)	Slow Eating (20 min)
Mean plasma glucose concentration (mmol/L)	4.76 ± 0.11	4.79 ± 0.12
SD of plasma glucose concentration (mmol/L)	1.18 ± 0.10 *	0.92 ± 0.06
MAGE (mmol/L)	3.67 ± 0.31 **	2.67 ± 0.20
IGP after breakfast (mmol/L)	2.30 ± 0.19 **	1.71 ± 0.12
IGP after lunch (mmol/L)	4.06 ± 0.33 **	3.13 ± 0.28
IGP after dinner (mmol/L)	3.87 ± 0.38 ***	2.27 ± 0.27
IAUC for glucose of breakfast 0–120 min (mmol/L × min)	111 ± 10	107 ± 8
IAUC for glucose of lunch 0–120 min (mmol/L × min)	265 ± 28	216 ± 21
IAUC for glucose of dinner 0–120 min (mmol/L × min)	256 ± 30 ***	128 ± 18

Data are mean ± SEM. SD—standard deviation of plasma glucose concentration; MAGE—mean amplitude of glycemic excursion; IGP—incremental glucose peak; IAUC—incremental area under the curve. The mean plasma glucose, SD, and MAGE were calculated from 7:00 to 7:00 in the following day. The IAUCs for glucose of each meal were calculated by the trapezoidal method. * *p* < 0.05, ** *p* < 0.01, *** *p* < 0.001.

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
