# Peer review of "Eating Fast Has a Significant Impact on Glycemic Excursion in Healthy Women: Randomized Controlled Cross-Over Trial"

_nutrients, 2020, doi:10.3390/nu12092767_

Round 1
Reviewer 1 Report
This article subject is interesting as it shows the association between eating fast and glycemic excursion in normal healthy women. A few major points that need to be considered or clarified: 1) The methods section does not specify the criteria for healthy women or any information regarding their weight, BMI and age. All of these are important in understanding the population. 2) The meals were eaten at home and not in a facility so adherence to time and eating pattern is self reported. It cannot be managed through phone calls between participant and researchers. How was adherence determined? 3) Why were the same individuals put on the two different eating speed diet? 4)The study would have been much stronger with measurement of satiety measurements. 5) The discussion does not go into details of what makes this study important to the field and why it is novel.
Author Response
Reviewer 1
Comments and Suggestions for Authors
This article subject is interesting as it shows the association between eating fast and glycemic excursion in normal healthy women. A few major points that need to be considered or clarified:
1) The methods section does not specify the criteria for healthy women or any information regarding their weight, BMI and age. All of these are important in understanding the population.
Response
Thank you for your valuable comments. We rewrote the characteristic of the participants in 2.1 Participants and Results.sections described as below.
P2 L55-59
2.1. Participants
University students were recruited from Kyoto Women’s University, Kyoto, Japan and after being informed the study requirements 21 volunteered to participate. The study was conducted between Dec. 2019 and Feb. 2020. The volunteers had no history of any metabolic diseases. None of the volunteers were pregnant, smokers, had eating disorder, weight loss, or had other special diet in the previous 6 months, and refrained from taking any medications and supplements known to affect their metabolism. The volunteers had no history of any metabolic diseases. …
Results
Therefore, the results were based on 19 women [20.8 ± 0.6 years, BMI 20.6 ± 1.9 kg/m2, HbA1c 34 ± 2 mmol/mol (5.4 ± 0.2%), FPG 4.86± 0.39 mmol/L: mean ± SD].
2) The meals were eaten at home and not in a facility so adherence to time and eating pattern is self reported. It cannot be managed through phone calls between participant and researchers. How was adherence determined?
Response
Thank you for important comment. As you say, the meals were eaten at home and not in a facility so adherence to time and eating pattern is self-reported. However, in this study, to increase the adherence, each participant was instructed to follow the study protocol precisely during the study period and the collected records of eating speed and the amounts of food were assessed for compliance of the study protocol by the dietitians of the study group. In addition, the dietitians excluded the participants who did not follow the protocol and the phone calls were made to participants by the researchers to maintain the study protocol. Thus, we believe that each participant understood the study meaning and adhered the study protocol. We have added the sentences in the study protocol and mentioned the test meals as follows and the Results. However, there is a possibility of not adhering the study protocol. Thus, we have mentioned this point as one of the limitations of this study described as below.
P3 L101-104
2.3. Test meals
Each participant was instructed to follow the study protocol precisely during the study period and the collected records of eating speed and the amounts of food were assessed for compliance of the study protocol by the dietitians of the study group. The dietitians excluded the participants who did not follow the protocol.
P2 L63-
2.2. Study Design
This study was a randomized controlled two-treatment cross-over within-participant clinical trial. The study protocol involving human subjects was approved by the Ethics Committee of Kyoto Women’s University (2019-8) according to the guideline laid down in the Declaration of Helsinki and registered at UMIN Clinical Trials Registry (UMIN 0000038684).
All participants consumed identical test meals for two days which were consumed at two different eating speed during the 6 days study period as shown in Figure. 1. The study protocol was explained to each participant prior and during the study, and the phone calls were made to participants by the researchers to maintain the study protocol. The participants wore FGMs on the back of their left upper arm for 6 days under the physician’s management at Kyoto Women’s University. FGM system starts working 1 hour from application, although, we started to assess glucose parameters 3 days after wearing it for accuracy of glycemic parameters. On the 4th day, participants were instructed to consume three meals either quickly (10 minutes) or slowly (20 minutes). On the 5th day, they were instructed to consume each their meals at the opposite speed.
P4 L129-
Results
Twenty one participants were enrolled in the study and 2 participants were excluded because of the discontinuation of the study protocol of eating speed.
Discussion
P7 L200- Sixth, there is a possibility of not adhering the study protocol, although each participant understood the study meaning.
3) Why were the same individuals put on the two different eating speed diet?
Response
Thank you for your comment. We designed this study as randomized controlled cross-over trial, since we can avoid the characteristic difference of two groups in cross-over trial. We performed a paired comparison between fast and slow eating within the same participant by Wilcoxon matched-pairs signed-rank test.
4)The study would have been much stronger with measurement of satiety measurements.
Response
Thank you for your important comment. We did not assess the satiety measurements in this study, so we have added the fifth limitation of the study in Discussion.
P6L198-
Fifth, we did not measure the satiety difference between fast and slow eating. Because, eating fast has been reported to increase the excess food intake since eating fast lead to lower satiety [11-13].
5) The discussion does not go into details of what makes this study important to the field and
why it is novel.
Response
Thank you for your valuable comment. According to your comment, we have mentioned in Discussion as follows and we rewrote the conclusion.
P5L149-
To the best of our knowledge, this is the first interventional study to investigate the association between eating speed and glycemic parameters by flash glucose monitoring system. The results of this study suggest that the fast eating is associated with higher postprandial glucose concentrations and daily glycemic excursions in young healthy women…. However, eating speed of these reports was not assessed by interventional study, the eating fast was obtained from questionnaires by participants themselves might yield a report bias and the rate of eating speed has not been measured objectively.
P7 L204-
The disadvantage of eating fast shown in this study brings the possibility to raise the risk of type 2 diabetes and obesity in healthy young individuals. Eating slowly are potentially advantage of public health, because the modification of eating speed could be a cost effective for promoting management of body weight and glycemic control [33, 34]. The dietary education of benefit of eating slow might be a simple way to reduce excess food intake since eating fast lead to higher energy intake but lower satiety [11-13]. The real-world plans for slow eating are to perform individuals in lunch break at school or working place. However, in future studies, additional investigations are required to explain the mechanisms under this effects and long-term effects of eating speed on metabolic control in individuals with and without diabetes.
Conclusion
To summarize, we have shown that the eating fast is associated with higher daily glycemic excursions and postprandial glucose levels for the first time in randomized controlled cross-over trial for the first time. Real-world approaches are needed to better understanding of the negative influence of eating fast on glycemic responses and to support approaches for slowing down the eating speed in healthy individuals.
Submission Date
11 August 2020
Date of this review
18 Aug 2020 21:07:34

Reviewer 2 Report
1 - At the beginning of the article should be assumed and clarified that the analysis is about the acute effect on glycemia. The authors assume that during the limitation part, however, it's important to clarify the readers at the beginning of the paper.
2 (test meals section) - Are these meals normally consumed by this population? Can this type of meal (because are very different from the habitual consumed) influence glycemia? Maybe the authors should assume it as a limitation.
3 (Results section, line 124-126) - This first sentence belongs to the methods section and not the results section. Please correct it.
Author Response
Reviewer 2
フォームの始まり
Comments and Suggestions for Authors
This is an interesting piece of novel research, which appears to have been well executed and presented.
We really appreciate your valuable comments and suggestions. We corrected and added the sentences with yellow marker in the manuscript.
- Suggest changing 'parameters in randomized' to 'parameters through conducting a randomized'
25 & 27. Write out numbers under ten in full
Response
Thank you for your valuable comment. We corrected them.
P1 L23-
Our aim was to evaluate the effect of fast eating on glycemic parameters through conducting a randomized controlled…
- The three referenced articles do not conclusively demonstrate a link between speed of eating and weight gain. I would suggest this sentence is revised to reflect this. The three studies also use self-reported eating speeds, which is not clear as this is only mention in the following sentences.
Response
Thank you for your valuable comments. We excluded the reference 1 which does not demonstrate a link between eating speed and weight gain and rewrote the introduction as follows.
P1 L38-
It has been demonstrated that self-reported eating fast speed leads to weight gain [1, 2], increases risk of type 2 diabetes [3-5], obesity [6, 7], and metabolic syndrome [8-10] in epidemiological and cohort studies. Eating fast encourages higher energy intake by increasing food stimuli, hunger, and the desire to eat [11-13].
- Speeds
- End sentence after women
48-50. Rewrite sentence (suggestion - Flash glucose monitoring system (FGM....) were utilized. FGM has been shown to be an accurate etc.). It may also be beneficial to have a sentence explaining how the FGM works and was utilised for this study.
Response
Thank you for your valuable comment. We added the sentences how the FGM worked and why it was used for this study.
P2 L47-53
The aim of this study was to evaluate the acute effect of different eating speed on glycemic parameters in young healthy women with flash glucose monitoring system (FGM, FreeStyle Libre Pro, Abbott Japan, Tokyo). FGM is a new technology of continuous glucose monitoring system that does not require regular capillary glucose sampling by finger prick (self-monitored blood glucose, SMBG) like a traditional continuous glucose monitoring (CGM). FGM sensor stores interstitial fluid glucose levels every 15 minutes for the 14 days and was reported to be accurate and effectively replaced SMBG [14-16].
University students from Kyoto... were recruited, and after being informed of the study requirements, 21 volunteered to participate.
- had eating disorders,
Response
Thank you. We corrected them.
2.1. Participants
P2 L56-59
University students were recruited from Kyoto Women’s University, Kyoto, Japan and after being informed the study requirements 21 volunteered to participate. The study was conducted between Dec. 2019 and Feb. 2020. The volunteers had no history of any metabolic diseases. None of the volunteers were pregnant, smokers, had eating disorder,,…
Study design - not clear why the participants wore the FGM for six days but data were only collected/presented for the two test meal days. Maybe add explanation.
Response
Thank you for your important point. We added the following sentences
P2 L73-76
FGM system starts working 1 hour from application, although, we started to assess glucose parameters 3 days after wearing it for accuracy of glycemic parameters. On the 4th day, participants were instructed to consume three meals either quickly (10 minutes) or slowly (20 minutes). On the 5th day, they were instructed to consume each their meals at the opposite speed.
rather than 'in two different eating speeds' maybe 'which were consumed at two different speeds'
- participants wore FGMs on the back of their left
- and phone calls were made to the participants by the researchers
- Probably needs more explanation i.e. On day four, participants were asked to eat each of their three meals either quickly (over 10 minutes) or slowly (over 20 minutes). On day five, they were required to each their meals at the opposite speed.
- homogeneity for all
Response
Thank you for your important comments. We corrected them as follows.
P2 L68-76
All participants consumed identical test meals for two days which were consumed at two different eating speed during the 6 days study period as shown in Figure. 1. The study protocol was explained to each participant prior and during the study, and the phone calls were made to participants by the researchers to maintain the study protocol. The participants wore FGMs on the back of their left upper arm for 6 days under the physician’s management at Kyoto Women’s University. FGM system starts working 1 hour from application, although, we started to assess glucose parameters 3 days after wearing it for accuracy of glycemic parameters. On the 4th day, participants asked to consume three meals either quickly (10 minutes) or slowly (20 minutes). On the 5th day, they were required to consume each their meals at the opposite speed.
- I am unsure about the accuracy of the nutrient composition of the test meals. The breakfast states that it has 18.3g of protein and 12g of fat but it is unclear from the detail of the meal where this has come from. It is also not clear how the dinner has almost twice as much fat as the lunch, which contained fried fish. If there are missing foods, these need to be added. There are no quantities given for the fish and steak. ‘Gluten-meat’ doesn’t mean anything to me – is this a standard term?
Response
We corrected the nutrient composition of the test meals and enclosed the detail of the nutrient composition of the test meals as a supplemental table if it is necessary. We designed the three test meals according to the meal plan of Japanese dietary habit, which was 25% energy ratio of breakfast, 35%. energy ratio of lunch, and 40% energy ratio of dinner, because the results of this study should be easier to be applied to the real-world management. But your comment is very important, so we added the sentences the reason why only IAUC of dinner of fast eating demonstrated higher than those of slow eating of breakfast and lunch in discussion.
‘Gluten-meat’ steak is like a hamburger steak for vegetarian. We insert the meal pictures in Figure 2 to show the detail of the test meals.
- References 1-3 do not link eating fast with over-overeating
Response
We corrected the references and rewrote the Discussion as follows.
P5 L 152-
Numerous studies reported that eating fast was associated with increased body weight and overeating [1, 2, 6, 7, 11-13, 20], elevated blood pressure and fasting plasma glucose concentration [8, 10], increased insulin resistance [4], and increased the risk of impaired glucose tolerance and type 2 diabetes [3, 5]. Other reports demonstrated the association between eating fast and lipid abnormality, such as elevated plasma triglyceride concentration and reduced plasma HDL concentration [10, 21].
P6 L174-
Additionally, another possible reason was that the energy ratio of dinner was large (40%) compared to that of breakfast (25%), and of lunch (35%). We employed this energy ratio to design the three test meals according to the general meal plans of Japanese dietary habit.
glucose intolerance
- interleukin-6, which are both
Response
Thank you, we corrected them.
P6 L161- Several reasons may explain the association of eating fast with a high risk of glucose intolerance and obesity.
P6 L164 that interleukin-1β and interleukin-6, which were both involved…
- Reference 25 only demonstrated differences in ghrelin and peptide YY
Response
Thank you for your valuable comment, we corrected them.
P6 L165-
Moreover, eating slowly may influence gastrointestinal satiety hormones, such as ghrelin and peptide tyrosine-tyrosine (PYY) which control appetite and influence food consumption, suggesting that modifiable eating behaviors actually regulate the hormonal response to food [25].
- reference written incorrectly
Response
Thank you for your important comment, we corrected the sentences and references as follow.
P5 L152-
Numerous studies reported that eating fast was associated with increased body weight and overeating [1, 2, 6, 7, 11-13, 20], elevated blood pressure and fasting plasma glucose concentration [8, 10], increased insulin resistance [4], and increased the risk of impaired glucose tolerance and type 2 diabetes [3, 5]. Other reports demonstrated the association between eating fast and lipid abnormality, such as elevated plasma triglyceride concentration and reduced plasma HDL concentration [10, 21].
- what is the relevance of eating speed in the daytime?
Response
We could not answer the relevance of eating speed in the daytime, but in the reference 26 they compared between eating breakfast slowly (30 min) and quickly (10 min). In our previous studies the participants consumed the similar test meal over 15 minutes (reference17, doi: 10.1016/j.diabres.2017.05.010, doi: 10.1016/j.diabet.2018.07.001, doi: 10.1016/j.diabres.2017.11.033. doi: 10.1016/j.diabet.2018.10.004. doi: 10.6133/apjcn.202003).
Therefore, the result of this study might be one of the evidence of the relevance of eating speed in the daytime.
be verified
platelet - something missing. Maybe should be plural or platelet aggregability?
Response
Thank you we corrected it.
P6 L179 …to be verified in further studies.
P6 L185 and platelet aggregability [31]
185-188. people don’t consume ‘carbohydrate’ they consume foods with a high carbohydrate content
Response
Thank you, we corrected them.
P6 L196- …each meal was consumed by meal sequence of vegetable followed by main dish, and rice/bread, while in fast eating all meals were consumed as mixture of vegetable, the main dish, and rice/bread .
review sentence structure starting Our findings…
Response
Thank you, we corrected them.
P7 L205-
Eating slowly are potentially advantage of public health, because the modification of eating speed could be a cost effective for promoting management of body weight and glycemic control [33, 34]
- A single study isn’t sufficient evidence to claim that ‘eating fast lead to higher energy intake but lower satiety’. It could support that this might be the case.
Response
We rewrote the following sentences.
P7 L207-
The dietary education of benefit of eating slow might be a simple way to reduce excess food intake since eating fast lead to higher energy intake but lower satiety [11-13].
- Sentence starting “The practical strategies’ needs rewording.
Response
Thank you for your valuable comment. We wrote the sentence and the conclusions as follows.
P7 L209
The real-world approaches for slow eating are to perform individuals in lunch break at school or working place….
P7 L213- Conclusions
To summarize, we have shown that the eating fast is associated with higher daily glycemic excursions and postprandial glucose levels for the first time in randomized controlled cross-over trial for the first time. Real-world approaches are needed to better understanding of the negative influence of eating fast on glycemic responses and to support approaches for slowing down the eating speed in healthy individuals.
General; How was the speed of eating determined i.e. how was the 10-minute time-frame established as ‘fast’ eating and the 20-minutes as ‘slow’? Did the size and composition of the meal not determine what was fast and slow eating (i.e. the smaller breakfast, with nothing that required high levels of chewing vs. the steak dinner)? Were the participants asked about their normal eating speeds and then asked to assess whether they felt they were eating quickly or slowly? It would be have more detail in the methodology to explain why these parameters were chosen and tested.
Response
Thank you for your valuable comment. The participants were measured eating time to consume the test meals in normal eating speed and in very fast eating speed in the preliminary test. Ten minutes is the minimum eating time for young Japanese women to consume the test meals. We decided the 20 min as a slow eating because the similar test meals were consumed 15 minutes in our previous studies (reference17, doi: 10.1016/j.diabres.2017.05.010, doi: 10.1016/j.diabet.2018.07.001, doi: 10.1016/j.diabres.2017.11.033. doi: 10.1016/j.diabet.2018.10.004. doi: 10.6133/apjcn.202003)
Submission Date
11 August 2020
Date of this review
20 Aug 2020 03:08:24
フォームの終わり

Reviewer 3 Report
This is an interesting piece of novel research, which appears to have been well executed and presented.
- Suggest changing 'parameters in randomized' to 'parameters through conducting a randomized'
25 & 27. Write out numbers under ten in full
- The three referenced articles do not conclusively demonstrate a link between speed of eating and weight gain. I would suggest this sentence is revised to reflect this. The three studies also use self-reported eating speeds, which is not clear as this is only mention in the following sentences.
- speeds
- End sentence after women
48-50. Rewrite sentence (suggestion - Flash glucose monitoring system (FGM....) were utilized. FGM has been shown to be an accurate etc.). It may also be beneficial to have a sentence explaining how the FGM works and was utilised for this study.
- University students from Kyoto... were recruited, and after being informed of the study requirements, 21 volunteered to participate.
- had eating disorders,
Study design - not clear why the participants wore the FGM for six days but data were only collected/presented for the two test meal days. Maybe add explanation.
- rather than 'in two different eating speeds' maybe 'which were consumed at two different speeds'
- and phone calls were made to the participants by the researchers
- participants wore FGMs on the back of their left
- Probably needs more explanation i.e. On day four, participants were asked to eat each of their three meals either quickly (over 10 minutes) or slowly (over 20 minutes). On day five, they were required to each their meals at the opposite speed.
- homogeneity for all
- I am unsure about the accuracy of the nutrient composition of the test meals. The breakfast states that it has 18.3g of protein and 12g of fat but it is unclear from the detail of the meal where this has come from. It is also not clear how the dinner has almost twice as much fat as the lunch, which contained fried fish. If there are missing foods, these need to be added. There are no quantities given for the fish and steak. ‘Gluten-meat’ doesn’t mean anything to me – is this a standard term?
- References 1-3 do not link eating fast with over-overeating
- glucose intolerance
- interleukin-6, which are both
- Reference 25 only demonstrated differences in ghrelin and peptide YY
- reference written incorrectly
- what is the relevance of eating speed in the daytime?
- be verified
- platelet - something missing. Maybe should be plural or platelet aggregability?
185-188. people don’t consume ‘carbohydrate’ they consume foods with a high carbohydrate content
- review sentence structure starting Our findings…
- A single study isn’t sufficient evidence to claim that ‘eating fast lead to higher energy intake but lower satiety’. It could support that this might be the case.
- Sentence starting “The practical strategies’ needs rewording.
General; How was the speed of eating determined i.e. how was the 10-minute time-frame established as ‘fast’ eating and the 20-minutes as ‘slow’? Did the size and composition of the meal not determine what was fast and slow eating (i.e. the smaller breakfast, with nothing that required high levels of chewing vs. the steak dinner)? Were the participants asked about their normal eating speeds and then asked to assess whether they felt they were eating quickly or slowly? It would be have more detail in the methodology to explain why these parameters were chosen and tested.
Author Response
This is a comment (supplemental table) to the reviewer 2.

Round 2
Reviewer 1 Report
Minor revision: It is crucial to specify why the same individuals were used to test both eating speeds. I would suggest including the reasoning mentioned in your response
"We designed this study as randomized controlled cross-over trial, since we can avoid the characteristic difference of two groups in cross-over trial. We performed a paired comparison between fast and slow eating within the same participant by Wilcoxon matched pairs signed-rank test. "
Author Response
Reviewer 1
Minor revision: It is crucial to specify why the same individuals were used to test both eating speeds. I would suggest including the reasoning mentioned in your response
"We designed this study as randomized controlled cross-over trial, since we can avoid the characteristic difference of two groups in cross-over trial. We performed a paired comparison between fast and slow eating within the same participant by Wilcoxon matched pairs signed-rank test. "
Response
Thank you for your valuable comments. We added the sentences in 2.2. Study design as below. We have mentioned the Statistical Analysis in
P2 L64-
This study was a randomized controlled two-treatment cross-over within-participant clinical trial, since we could avoid the characteristic differences of two groups in different eating speed.
P4 L123-
2.5. Sample Size and Statistical Analysis
…. so we performed a paired comparison by Wilcoxon matched-pairs signed-rank test, and p < 0.05 was considered statistically significant. The results are expressed as mean ± standard error of the mean (SEM) unless otherwise stated. All analyses were performed with SPSS Statistics ver. 22 software (IBM Corp., Armonk, NY, USA).